

# Inverse modeling of Chinese NOx emissions using deep learning: Integrating in situ observations with a satellite-based chemical reanalysis

Tai-Long He[1], Dylan B. A. Jones[1], Kazuyuki Miyazaki[2], Kevin W. Bowman[2], Zhe Jiang[3], Xiaokang Chen[3], Rui Li[3], Yuxiang Zhang[3], and Kunna Li[1]

[1]Department of Physics, University of Toronto, Toronto, Ontario, Canada
[2]Jet Propulsion Laboratory, California Institute of Technology, Pasadena, CA, USA
[3]School of Earth and Space Sciences, University of Science and Technology of China

**Correspondence:** Tai-Long He (tailong.he@mail.utoronto.ca)

**Abstract.** Nitrogen dioxide ($NO_2$) column density measurements from satellites have been widely used in constraining emissions of nitrogen oxides ($NO_x$ = NO + $NO_2$). However, the utility of these measurements is impacted by reduced observational coverage due to cloud cover and by their reduced sensitivity toward the surface. Combining the information from satellites with surface observations of $NO_2$ will provide greater constraints on $NO_x$ emission estimates. We have developed a deep learning (DL) model to integrate satellite data and in situ observations of surface $NO_2$ to estimate $NO_x$ emissions in China. A prior information for the DL model was obtained from satellite-derived emissions from the Tropospheric Chemistry Reanalysis (TCR-2). A two-stage training strategy was used to integrate in situ measurements from the China Ministry of Ecology and Environment (MEE) observation network with the TCR-2 data. The DL model is trained from 2005 to 2018, and is evaluated for 2019 and 2020. The DL model estimated a source of 19.4 Tg NO for total Chinese $NO_x$ emissions in 2019, which is consistent with the TCR-2 estimate of 18.5±3.9 Tg NO and the 20.9 Tg NO suggested by the Multi-resolution Emission Inventory for China (MEIC). Combining the MEE data with TCR-2, the DL model suggested higher $NO_x$ emissions in some of the less densely populated provinces, such as Shaanxi and Sichuan, where the MEE data indicated higher surface $NO_2$ concentrations than TCR-2. The DL model also suggested a faster recovery of $NO_x$ emissions than TCR-2 after the Chinese New Year (CNY) holiday in 2019, with a recovery time scale that is consistent with Baidu "Qianxi" mobility data. In 2020, the DL-based analysis estimated about a 30% reduction in $NO_x$ emissions in eastern China during the COVID-19 lockdown period, relative to pre-lockdown levels. In particular, the maximum emission reductions were 42% and 30% for the Jing-Jin-Ji and the Yangtze River Delta megaregions, respectively. Our results illustrate the potential utility of the DL model as a complementary tool for conventional data assimilation approaches for air quality applications.

## 1 Introduction

Nitrogen oxides ($NO_x$ = NO + $NO_2$) are a family of primary air pollutants that are directly involved in the formation of other air pollutants, such as tropospheric ozone and secondary inorganic aerosols. $NO_x$ is emitted by anthropogenic sources





on the surface, and it also has natural sources from lightning in the free troposphere (Murray, 2016). Satellite observations of tropospheric $NO_2$ column have been widely used during the past two decades to constrain $NO_x$ emissions (referred to as "top-down" emissions). Martin et al. (2003) used a mass balance approach with the GEOS-Chem global chemical transport

model (CTM) to relate changes the $NO_2$ column to $NO_x$ emissions at the surface. They showed that the top-down analysis could reduce regional uncertainties in the a priori $NO_x$ emissions. Satellite-derived $NO_x$ emissions have been obtained by several subsequent studies using a similar mass balance approach (Bertram et al., 2005; Konovalov et al., 2006; Kim et al., 2006; Martin et al., 2006; Toenges-Schuller et al., 2006; Boersma et al., 2008). Advanced data assimilation methods have also been applied to obtain satellite-based emission estimates of $NO_x$. For example, the four-dimensional variational (4D-Var)

method uses a CTM and its adjoint to propagate the differences between satellite data and simulation to the a priori estimate of $NO_x$ emissions (Müller and Stavrakou, 2005; Kurokawa et al., 2009; Chai et al., 2009; Qu et al., 2019). The Kalman filter is another widely used method, which employs information about the error covariance in the forecast of the trace gases to update atmospheric quantities in the CTMs (Napelenok et al., 2008; Miyazaki et al., 2020a; Wu et al., 2020).

Despite the range of inverse modeling approaches used to estimate $NO_x$ emissions from satellite observations, they all
suffer from potential limitations associated with the chemical transport models (CTMs) employed in the inversion analyses. For example, Lin and McElroy (2010) found that a different scheme for mixing in the planetary boundary layer could lead to 3–14% differences in the top-down $NO_x$ emission budgets for East China. Deep convective transport in the free troposphere, which can be challenging to accurately simulate, could vertically transport $NO_2$ generated by lightning activities, which results in greater non-linearity between $NO_x$ emissions and $NO_2$ columns (Choi et al., 2005; Nault et al., 2017). The lifetime of
$NO_x$ varies diurnally and seasonally, and discrepancies in the ability of a CTM to capture these variations will contribute to uncertainties in the top-down $NO_x$ emission estimates (Beirle et al., 2011; de Foy et al., 2014; Liu et al., 2016).

An additional limitation with the satellite-based top-down emission estimates of $NO_x$ is that $NO_2$ is highly concentrated near the surface, whereas satellite measurements have lower sensitivity near the surface (Boersma et al., 2016). As a result, it is challenging for satellite observations to capture changes in surface $NO_x$ emissions. It has been suggested that satellite
$NO_2$ measurements are blending information from both surface emissions and atmospheric background of $NO_2$ due to the low sensitivity of the satellite retrievals to $NO_2$ near the surface (Li and Wang, 2019; Silvern et al., 2019; Qu et al., 2021). In situ observations of surface $NO_2$ are more representative of local emissions, but typically have more limited observational coverage. As a result, combining the surface observations with the satellite data could offer greater constraints on $NO_x$ emission estimates.

Here we use a deep learning (DL) model to indirectly integrate satellite data and in situ observations of surface $NO_2$ to estimate $NO_x$ emissions in China. Deep learning and other machine learning models have been increasingly used in the field of atmospheric science (Rasp et al., 2018; He et al., 2022; Keller and Evans, 2019). These data-driven methods show high skill in capturing nonlinear relationship between correlated atmospheric quantities. Compared to conventional data assimilation systems, DL models are free of errors in chemistry and the potential errors associated with defective parameterization
of subgrid-scale processes (Rasp et al., 2018). Moreover, high-resolution data assimilation using conventional approaches is computationally expensive especially when dealing with large amounts of data, whereas DL models show much higher compu-





tational efficiency for high-resolution data-rich applications. In the present work, we use a DL model to estimate Chinese $NO_x$ emissions using surface $NO_2$ concentrations. We use a two-stage transfer learning strategy to integrate in situ $NO_2$ observations from the China Ministry of Ecology and Environment (MEE) network with the Tropospheric Chemical Reanalysis (TCR-2)

that assimilated satellite observations.

    We focus on the 2019–2020 period, which overlaps with the COVID-19 pandemic that led to the lockdown of over one-third of Chinese cities in early 2020. Observations have shown significant reductions of atmospheric abundances of $NO_2$ over China during this period (Bauwens et al., 2020; Liu et al., 2020). The change in atmospheric $NO_2$ implies an anomalous change in emission of $NO_x$, which provides a unique opportunity to evaluate the utility of the DL model for estimating $NO_x$ emissions.

We evaluate the performance of the DL model by analyzing the predicted $NO_x$ emissions for the normal year 2019 and the anomalous year 2020. Evaluation of the DL-based system utilized the dependent testing data set from the TCR-2 standard product (based on data from the Ozone Monitoring Instrument (OMI)), an updated higher-resolution TCR-2 reanalysis product constrained by the TROPOspheric Monitoring Instrument (TROPOMI) measurements, and the independent Baidu "Qianxi" mobile data (Kraemer et al., 2020; Zhang et al., 2021).

The outline of the paper is as follows. Section 2 describes the data sets used in the analysis, the DL model, and the two-stage training strategy. Section 3 shows the results from the evaluation of the model performance after the two training stages and the analyses of the Chinese $NO_x$ emissions for 2019 and 2020. Conclusions are presented in Section 4.





## 2 Data and methods

### 2.1 TCR-2 chemical reanalysis

The TCR-2 chemical reanalysis (Miyazaki et al., 2020a) was constructed using a Local Ensemble Transform Kalman Filter (LETKF) data assimilation system (Hunt et al., 2007), which optimizes both emissions and atmospheric abundance of various chemical species from assimilation of multi-constituent measurements from multiple satellite instruments. The observational data are assimilated into the MIROC-CHASER global chemical transport model (Sudo et al., 2002; Sekiya et al., 2018). The TCR-2 data product has a horizontal resolution of $1.1° \times 1.1°$, and consists of 27 pressure levels from 1000 to 60 hPa. Details
about the TCR-2 data assimilation system can be found in Miyazaki et al. (2020a).

The TCR-2 $NO_x$ emissions were constrained in part by the Ozone Monitoring Instrument (OMI) $NO_2$ measurements. OMI is a spectrometer on-board the NASA Aura spacecraft that was launched on 15 July 2004. It measures $NO_2$ in the UV-VIS range of the spectrum, from which vertical column densities (VCD) of $NO_2$ are retrieved. The OMI measurement strategy provides global coverage once per day. It should be noted that as a chemical data assimilation system with detailed tropospheric
chemistry, TCR-2 also relies on observational constraints from other $NO_2$-related chemical species (e.g. tropospheric ozone) to optimize tropospheric $NO_2$ and $NO_x$ emissions. For the analysis conducted here, the TCR-2 surface $NO_2$ concentrations and $NO_x$ emissions were averaged to daily fields to train the DL model. In addition to the standard TCR-2 product, we also used an updated version of the TCR-2 products that assimilated TROPOMI $NO_2$ data at a higher spatial resolution of $0.5625°$ (referred to as the T213 product Miyazaki et al. (2020b, 2021)). TROPOMI serves as the continuation and the next generation
of the OMI sensor, monitoring air pollutants at a much higher horizontal resolution (7 km $\times$ 3.5 km at nadir), and data have been available since February 2018. Since the TROPOMI-based TCR-2 T213 product is only available for the last few years of the analysis period considered here, we use it as an independent data set in the evaluation of the DL model for the year 2020.

### 2.2 MEE network

As part of the Chinese government's Air Pollution Control Action Plan, the Ministry of Ecology and Environment of the
People's Republic of China have been deploying ground-based stations to monitor air pollution since 2013. Surface $NO_2$ concentration measurements are archived at an hourly frequency, at more than 1500 ground-based stations as of 2019. $NO_2$ concentrations are measured and reported in micro-grams per cubic meter ($\mu$g m$^{-3}$). Special attention is given to the reference state for unit conversion. From 2013 to August 31, 2018, the reference state for in situ measurements was 273 K and 1013 hPa, after which it was changed to 298 K and 1013 hPa. To simplify the integration of the MEE data with the TCR-2 chemical
reanalysis, the in situ measurements were aggregated to the $1.1° \times 1.1°$ grid of the TCR-2 chemical reanalysis using the nearest neighbor interpolation method. The locations of the MEE sites as of 2019 are shown in Figure 1. The MEE network has good coverage over eastern and central China. For western China and northeastern China, the spatial coverage is lower. The MEE network includes both urban and rural sites and we included data from all sites to increase observational coverage and mitigate representation errors. The MEE measurements are made using a chemiluminescence analyzer with a molybdenum converter
that results in an overestimate of $NO_2$ (Lamsal et al., 2008). Following Lamsal et al. (2008), we used the GEOS-Chem model

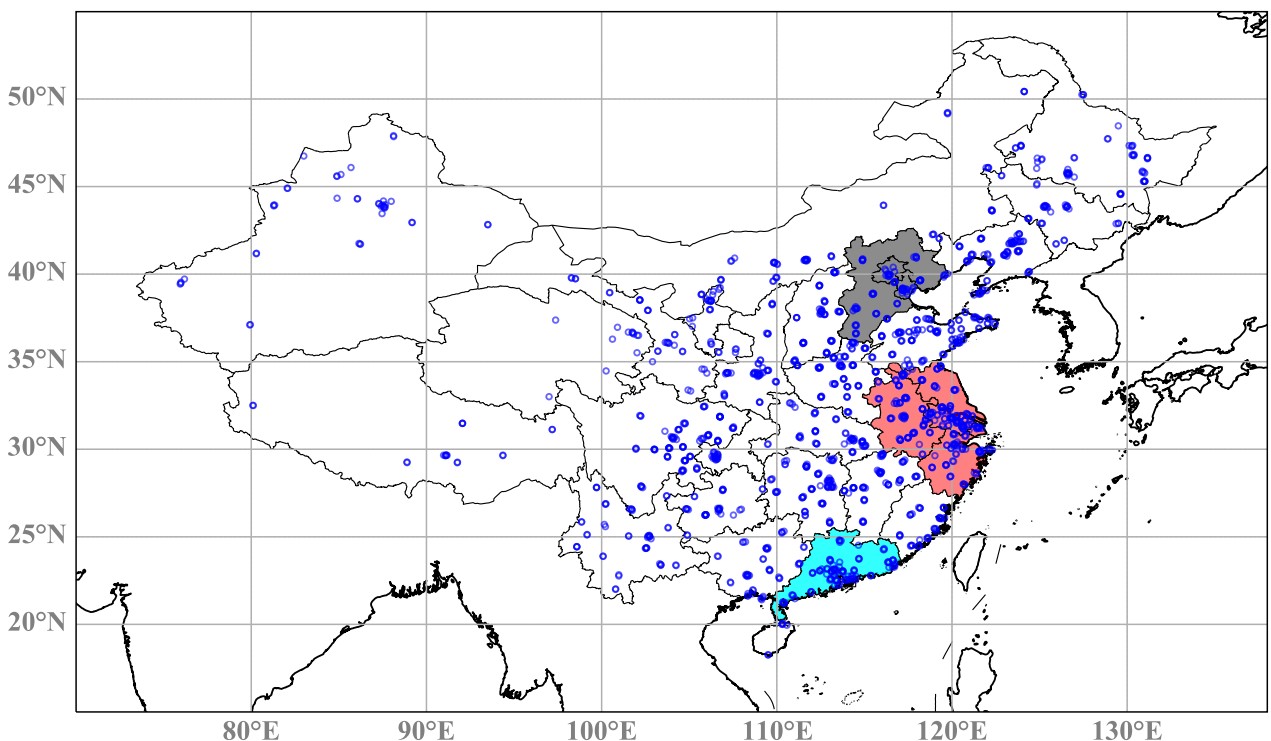

**Figure 1.** Location of the MEE network stations as of 2019. Blue circles represent the MEE stations. The 3 metropolitan regions that we focused on in the analysis are shaded in gray, red and cyan, for Jing-Jin-Ji (JJJ), Yangtze River Delta (YRD), and Pearl River Delta (PRD), respectively.

to simulate $NO_x$, $HNO_3$, peroxyacetyl nitrate (PAN), and alkyl nitrates (AN) to produce correction factors (CFs) for the measurements using the following relationship

$$CF = \frac{[NO_2]}{[NO_2] + 0.95[PAN] + 0.35[HNO_3] + \sum[AN]} \tag{1}$$

where $\sum AN$ is the sum of all ANs. We used version 12.0.2 of GEOS-Chem at a resolution of $2° \times 2.5°$ to estimate monthly mean CFs for 2015. The simulation was conducted with full chemistry and the MIX inventory (Li et al., 2017). The annual total Chinese emissions in the simulation was 19.0 Tg NO, which is consistent with TCR-2 estimates. These 2015 CFs were regridded to the $1.1° \times 1.1°$ resolution of the DL model and applied to the MEE measurements for 2014–2020. The CFs were close to unity in high emission regions, such as in eastern China, but could be as small as 0.4 in rural regions.



## 2.3 Baidu "Qianxi" mobile data

The Baidu "Qianxi" mobile data is generated from the Baidu map service, which is widely used in China as an equivalent to Google maps. It quantifies the intensity of human mobility as an immigration index (I-index), an emigration index (E-index), and an intra-city index (C-index). These migration indices have been used by other studies and are shown to have good correlation with human activities (Wei and Wang, 2020; Kraemer et al., 2020; Zhang et al., 2021). Zhang et al. (2021) found that the C-index, which gauged population flow inside cities, showed much higher correlation with variations in surface $NO_2$

concentration than the other two indices (the I-index and the E-index). We therefore use the C-index here as the independent data set to evaluate the DL model.

## 2.4 Deep learning model and input variables

The DL model used here is built using convolutional neural networks (CNNs) (Goodfellow et al., 2016) and long short-term memory (LSTM) units (Hochreiter and Schmidhuber, 1997). Figure 2 shows a schematic diagram of the model. The hybrid

architecture was previously used for predicting summertime surface ozone concentrations in the United States in He et al. (2022). The DL model showed great predictive skill in capturing the nonlinear relationship between the predictors and the model output. This DL architecture is applied here for estimating Chinese $NO_x$ emissions using surface $NO_2$ concentrations and meteorological variables as input predictors. The input variables are forwarded to three sequential convolutional blocks. Each convolutional block consists of two CNN layers and one max pooling layer. Each CNN layer uses filters that are $3 \times 3$ in

size to apply convolutional operations with the data vectors and output so-called latent vectors. The softplus function is applied to activate the output from each CNN layer to increase the non-linearity of the DL architecture. The max pooling layers further condense the dimension of the latent vectors by taking the maximum values within $2 \times 2$ windows. The first three convolutional blocks were used as an encoder for the extraction of spatial features hidden in the input information. The output of the last convolutional block was a highly compressed latent vector, which was then forwarded to two long-short term memory (LSTM)

cells, which are recurrent neural networks (RNNs), with 1024 units to learn the dynamics. The LSTM cell is followed by three up-convolutional layers and seven convolutional layers. The up-convolutional layers use $2 \times 2$ convolutional filters to up-sample the latent vectors to high-resolution outputs. We applied residual learning connections (the arrows in Figure 2), which were used to capture the more directly relationship between input and output variables and to stabilize the performance of the U-net model (Li et al., 2018; He et al., 2015). We used the Adam optimizer for boosted optimization of the U-net model

(Kingma and Ba, 2017). The Dl model was run on the NVIDIA T4 tensor core graphics processing unit (GPU) on the Graham supercomputer of Compute Canada. During the training process, the convolutional filters, the weight matrices and bias vectors in the LSTM were iteratively optimized using a back-propagation algorithm. The initial learning rate was $1 \times 10^{-5}$ for both training stages, and the residual sum of squares was used as the loss function. Other hyper-parameters for the training of the model include the number of epochs, which was 250, and the batch size, which was 30.

The meteorological variables were taken from the ERA5 climate reanalysis data product, which is different from the ERA-Interim data product used in the TCR-2 data assimilation system. We chose the more recent ERA-5 product because of its





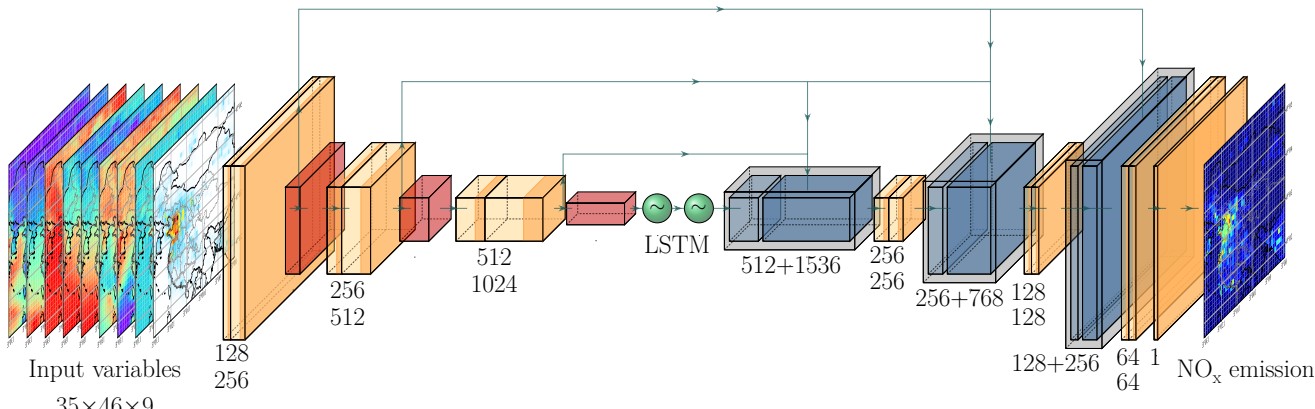

**Figure 2.** Schematic of the U-net model for the prediction of Chinese $NO_x$ emissions. The CNN layers with $3 \times 3$ filters are shown in orange. The dark orange regions indicate the application of softplus activation functions. The $2 \times 2$ max pooling layers are shown in red. The two green circles represent the LSTM cells. The $2 \times 2$ up-convolutional layers are shown in light blue, which are concatenated (indicated as grey boxes) with the transferred latent vectors (shown as dark blue boxes) from the encoder. The arrows indicate the residual connections.

higher spatial and temporal resolution, which better represents meso- to synoptic-scale transport processes (Hoffmann et al., 2019). This could be advantageous for the Stage 2 training, where the MEE in situ observations are used to improve the $NO_2$-$NO_x$ relationship. Table 1 lists the chosen input variables for the U-net model. All the input variables are regridded to

$1.1° \times 1.1°$ grid of TCR-2 using the first-order conservative remapping algorithm. As mentioned above, the output of the DL model, which is $NO_x$ emissions, is at the same $1.1° \times 1.1°$ grid. To ensure stability of the training of the DL model, all the input variables were scaled to make sure the values are within a relatively similar range. Table 1 gives the details about the input variables selected for the $NO_x$ emission inversions.

### 2.5 Two-stage training strategy

For the integration of the MEE in situ data with the TCR-2 reanalysis, we use a two-stage transfer learning strategy (Zhuang et al., 2020), as depicted in Figure 3. The first training stage focuses on the TCR-2 $NO_2$ concentration and $NO_x$ emission relationship. In this training stage, we train the model using the TCR-2 data pairs of $NO_2$ and $NO_x$, together with the ERA5 meteorological predictors. The purpose of this training stage is to supervise the DL model with the prior knowledge of the relationship between surface $NO_2$ concentrations and $NO_x$ emissions from the TCR-2 reanalysis. The goal here is to train the

DL model to reproduce the TCR-2 $NO_x$ emissions, given the TCR-2 surface $NO_2$ data. The second training stage is conducted based on the pre-trained DL model from Stage 1. This stage utilizes the transferred TCR-2 knowledge and the pre-trained DL model weights to provide an a priori for Stage 2. In this second stage, the MEE $NO_2$ data are integrated with the TCR-2 surface $NO_2$ data and the model retrained with the combined data set. The purpose of Stage 2 is to improve the relationship between surface $NO_2$ and $NO_x$ emissions acquired from TCR-2, given the available surface $NO_2$ observations. The TCR-2 reanalysis





**Table 1.** Input variables for $NO_x$ emission inversion using the DL model.

| Model input variables | Unit after scaling | Data source |
|---|---|---|
| Surface $NO_2$ concentration[1] | ppb | TCR-2/MEE[2] |
| Zonal component of 10-meter wind (U10m) | m s$^{-1}$ | ERA5 |
| Meridional component of 10-meter wind (V10m) | m s$^{-1}$ | ERA5 |
| 2-meter temperature (T2m) | K | ERA5 |
| Skin temperature (SKT) | K | ERA5 |
| Surface pressure (SP) | khPa | ERA5 |
| Shortwave radiation downwards (SRD) | kW m$^{-2}$ | ERA5 |
| Boundary layer height (BLH) | km | ERA5 |

[1] Surface $NO_2$ concentration fields include two time steps, for the current and previous days.

[2] TCR-2 surface $NO_2$ concentrations are used for the first training stage. The MEE in situ $NO_2$ measurements are added in the second training stage.

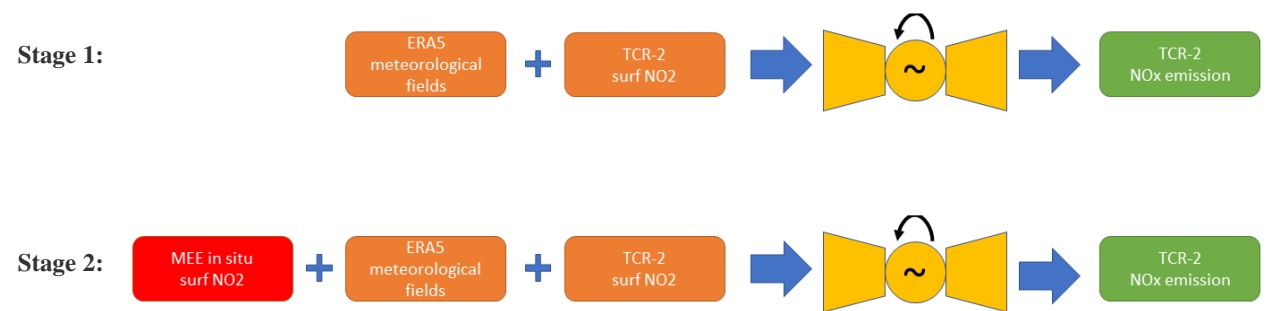

**Figure 3.** The two-stage training strategy used to integrate the in situ data with the TCR-2 chemical reanalysis. In Stage 1, only the ERA5 meteorological fields and the TCR-2 surface $NO_2$ data (represented by the orange boxes) are used as predictors in training the model (indicated by the yellow symbols) to predict $NO_x$ emissions (indicated by the green box). In Stage 2, the MEE in situ $NO_2$ measurements (denoted by the red box) are integrated with the TCR-2 data and the ERA5 meteorological fields as predictors.

spans from 2005 to 2020, whereas the MEE measurements are available beginning in 2014. Therefore, we train the DL model from 2005 to 2018 for Stage 1, and from 2014 to 2018 for Stage 2. The evaluation of the DL model is conducted for 2019 and 2020. Due to the impact of the COVID-19 pandemic, year 2020 is anomalous as compared to the training set with normal years. By including 2020 in the testing period, we examined the capability of the U-net model to extrapolate the training sample.



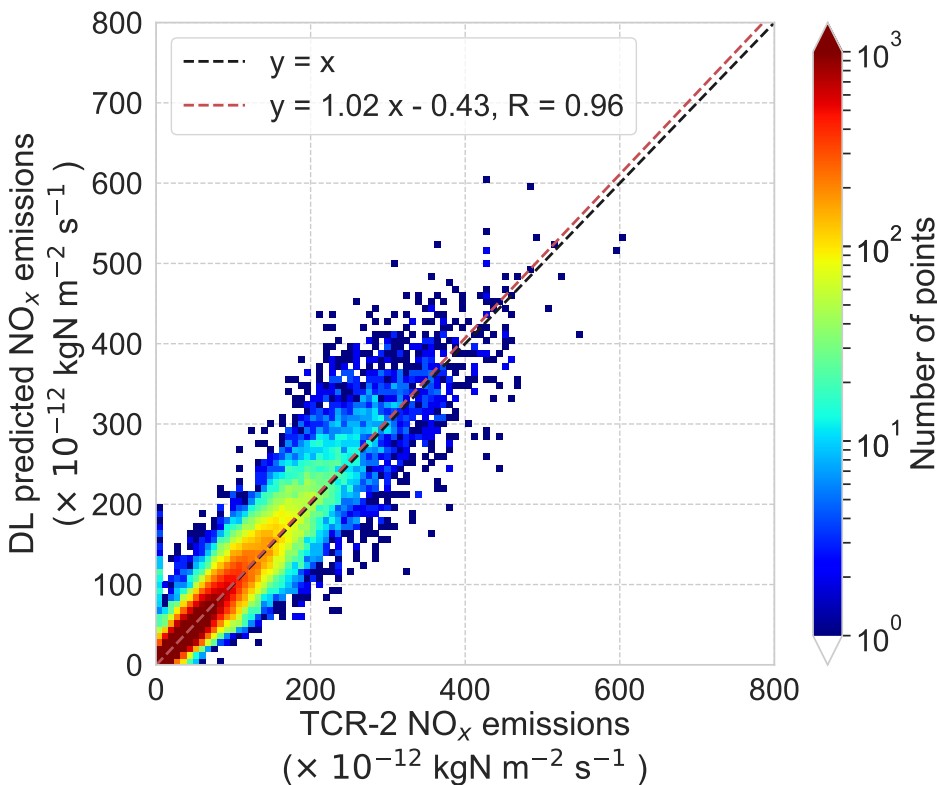

**Figure 4.** Correlation between daily NO$_x$ emissions from the TCR-2 reanalysis and the DL analysis for 2019 (the testing period).

## 3 Results and discussion

### 3.1 Analysis of the DL emission in 2019, during the testing period

A comparison of the Stage-1 DL analysis and the TCR-2 NO$_x$ emissions for 2019 is shown in Figure 4. The DL-estimated daily NO$_x$ emissions are in good agreement with the TCR-2 "truth" after the first training stage, with a correlation coefficient of 0.96 and a slope of 1.02. The annual mean errors in the DL-estimated NO$_x$ emissions are within 10%. These results indicate that the DL model captured well the relationship between surface NO$_2$ and NO$_x$ emissions from TCR-2 after the first training

stage. The time series of estimated NO$_x$ emissions for three metropolitan regions, five selected provinces, and four major cities in China are plotted in Figure 5. As can be seen, after Stage 1 the DL model agrees well with the TCR-2 emissions in most regions. Even on on small scales such as Beijing, which is only one grid box at the $1.1°$ resolution, the inferred emissions after Stage 1 are consistent with the TCR-2 emissions. The largest discrepancies after Stage 1 are found in the coastal regions of the Pearl River Delta (PRD) and the Yangtze River Delta (YRD), which encompass the cities of Guangzhou and Shanghai,

respectively, (see Figure 1 for the locations of the PRD and YRD regions).



The difficulty of the DL model in reproducing the TCR-2 emissions in the coastal regions, particularly in the the PRD, could reflect issues in both TCR-2 and the DL model. In the coastal regions in southern China, cloud cover will result in significantly reduced observational coverage from the satellites, which would impact TCR-2. For example, in the PRD (and Guangzhou) TCR-2 exhibits periods, such as around days 50 and 100, with anomalously low and constant $NO_x$ emissions, which could be due to reduced observational coverage. In addition, although both the PRD and the YRD experience heavy rainfall during the monsoon season, Luo et al. (2013) found that the PRD experience more frequent mesoscale convective systems and greater rainfall accumulation than the YRD, which they attributed to the mountainous topography of the PRD and the nearby ocean, in contrast to the more flat YRD. It is possible that at a resolution of $1.1° \times 1.1°$ both TCR-2 and the DL model cannot capture the complex meteorology (e.g., the sea breeze circulation and its model errors) in the PRD and its impact on the trace gas distribution, and thus are unable to reproduce the appropriate relationship between the $NO_x$ emissions and the atmospheric $NO_2$ concentrations.

After Stage 2, the $NO_x$ emissions calculated by the DL model are still consistent with the TCR-2 emissions in the main source emission regions. For Jing-Jin-Ji (JJJ), the DL emissions are 13.8% higher than the TCR-2 emissions, while for the YRD the DL emissions are 10.0% higher. The seasonal differences in the estimated emissions are given in Table 2. In JJJ, the differences between the DL and TCR-2 emissions are relatively constant throughout the year, whereas for the YRD the differences are small in the fall and larger in winter. In general, we find that the DL model suggest modest increases in emissions in central and eastern China, with relatively larger increases in the less densely populated provinces, such as Sichuan. In Sichuan, the estimated DL emissions are 23.2% higher than those in TCR-2 in summer. Comparison of the emissions after Stages 1 and 2 in Figure 5 shows that these large increases were produced after incorporating the MEE data in Stage 2. Thus, it is helpful to compare the time series of the TCR-2 and MEE surface $NO_2$ data, which are plotted in Figure 6. In Sichuan, the MEE observations suggest significantly higher surface $NO_2$ abundances, which could account for the higher DL emission estimates. For JJJ and the YRD, the TCR-2 $NO_2$ is in good agreement with the MEE data.

To further evaluate the estimated $NO_x$ emissions, in Figure 7 we compare the 2019 TRC-2 and DL emissions with the recently updated Multi-resolution Emission Inventory for China (MEIC) (Zheng et al., 2021). For total Chinese emissions, there is good consistency between the three inventories, with TCR-2, the DL model, and MEIC suggesting total Chinese emissions of 18.5±3.9, 19.4, and 20.9 Tg NO, respectively. However, despite the good agreement on the national scale, there are regional differences between the inventories. Compared to TCR-2, MEIC suggested higher $NO_x$ emissions in JJJ and in the Jiangsu province (the northeastern part of the YRD). The DL-estimated emissions are higher than TCR-2 in these regions, but lower than those of MEIC. In Inner Mongolia, both TCR-2 and the DL model infer higher emissions than MEIC, with the DL model suggesting more emissions than MEIC and less than TCR-2. For the PRD, TCR-2 $NO_x$ emissions are slightly lower than MEIC, whereas the DL results are slightly higher than MEIC. In less densely populated regions, such as Sichuan and Yunnan provinces the DL-estimated $NO_x$ emissions are higher than both TCR-2 and MEIC.





**Figure 5.** Time series of daily mean $NO_x$ emissions in 2019 for three Chinese metropolitan regions (Jing-Jin-Ji, the Yangtze River Delta, and the Pearl River Delta), five selected provinces (Henan, Shaanxi, Sichuan, Hubei, and Anhui), and four major cities (Beijing, Shanghai, Guangzhou, and Hefei) for 2019 (the testing period). Shown are the emissions from the TCR-2 reanalysis (black) and the DL analysis after Stage 1 (blue) and Stage 2 (red). The shaded areas represent the 14-day period after the Chinese New Year.

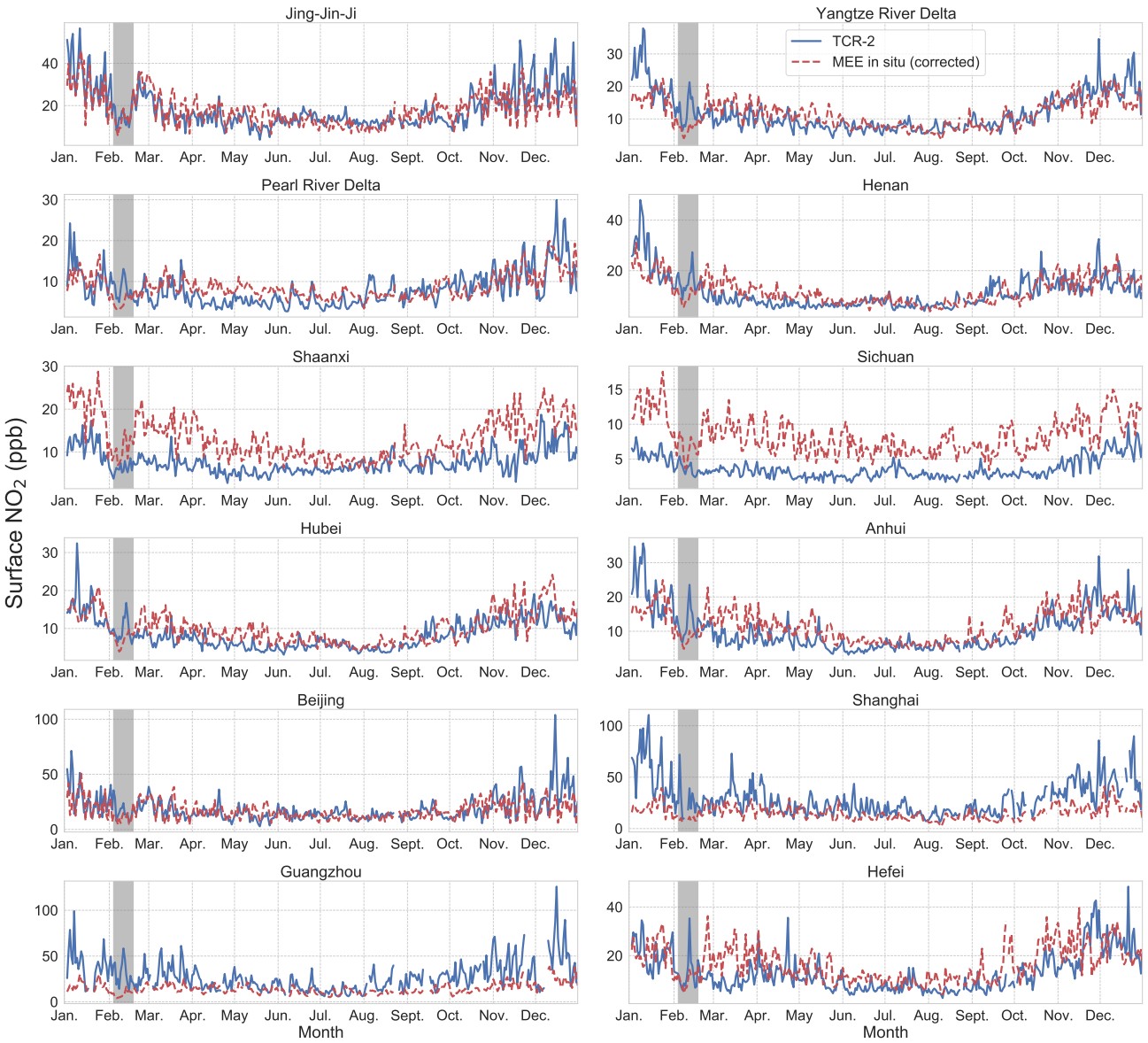

**Figure 6.** Time series of daily mean surface NO₂ concentrations in 2019 for the three Chinese metropolitan regions, selected provinces, and major cities shown in Figure 5. The corresponding TCR-2 NO₂ data sampled at MEE sites are shown in blue, whereas the MEE ground-based NO₂ observations are indicated by the dashed red line. The shaded areas represent the 14-day period after the Chinese New Year.

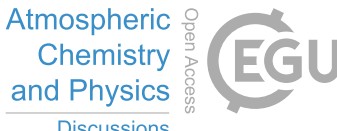

**Table 2.** Mean percentage difference between the estimated seasonal $NO_x$ emissions from the DL model (after Stage 2) and TCR-2 for 2019. Positive values represent that DL emissions are higher.

| Region/city | DJF | MAM | JJA | SON |
|---|---|---|---|---|
| China | 5.6 | 7.6 | 7.2 | 2.4 |
| Jing-Jin-Ji | 16.9 | 12.9 | 11.2 | 14.3 |
| Yangtze River Delta | 15.3 | 9.8 | 11.4 | 3.9 |
| Pearl River Delta | 12.3 | 34.3 | 21.5 | 0.5 |
| Henan | 8.5 | 4.9 | 14.4 | -1.8 |
| Shaanxi | 3.2 | 2.4 | 14.2 | 2.7 |
| Sichuan | -10.1 | 0.5 | 23.2 | 3.7 |
| Hubei | 12.7 | 7.6 | 25.6 | 7.3 |
| Anhui | 18.6 | 18.1 | 23.4 | 9.4 |
| Beijing | 9.1 | 19.1 | 54.8 | 11.1 |
| Shanghai | 5.0 | 2.3 | 20.0 | 8.8 |
| Guangzhou | 15.3 | 6.8 | 34.2 | 6.7 |
| Hefei | 36.3 | 16.4 | 33.3 | 19.0 |

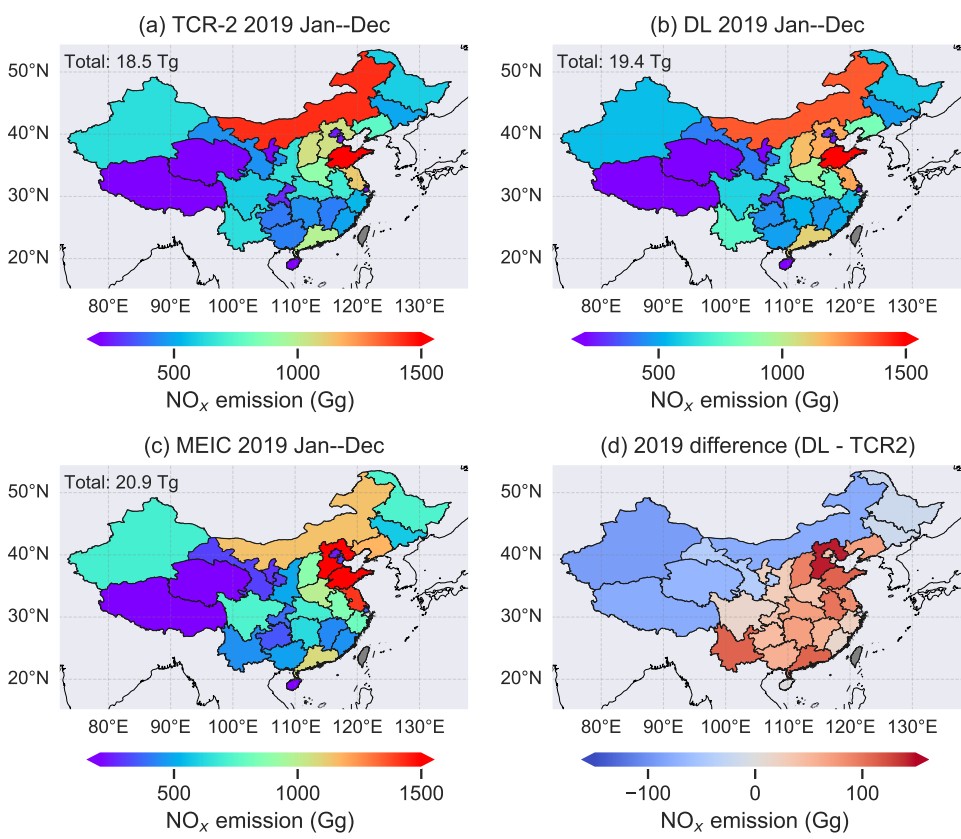

**Figure 7.** Annual NO$_x$ emissions (Tg NO) for 2019 estimated by (a) TCR-2 reanalysis, (b) the DL analysis and (c) the MEIC inventory. The DL minus TCR-2 emission differences are shown in (d).





## 3.2 Recovery of NO$_x$ emissions after the CNY holiday in 2019

Chinese NO$_x$ emissions typically decrease around January and February every year due to reduced human activity during
the CNY holidays. After the 1-week holiday, Chinese NO$_x$ emissions gradually recover to pre-holiday levels. This annual
variation in NO$_x$ emissions is well captured by the TCR-2 reanalysis, as shown in Figure 8. Starting from 10 days before
the holidays, the Chinese NO$_x$ emissions decrease rapidly by 20% relative to emissions 20 days before the CNY, reaching
a minimum shortly after CNY. The interannual spread in TCR-2 emissions was about ± 5% for the 2005–2019 period, as
similarly demonstrated by Miyazaki et al. (2020b), and the 2019 NO$_x$ emissions were consistent with the multi-annual mean.
The CNY-related variation in NO$_x$ emissions was captured in TCR-2 in all three of the megaregions, Jing-Jin-Ji, the Yangtze
River Delta, and the Pearl River Delta.

The DL analysis from Stage 2 was in good agreement with the TCR-2 reanalysis before the 2019 CNY for China and for
JJJ and YRD regions. However, after the holiday the DL-based NO$_x$ emissions recovered 50% of the post-holiday reductions
within 10–20 days, which was faster than in TCR-2 NO$_x$. The faster recovery in the DL-based NO$_x$ emissions can be clearly
seen for China and the YRD. The largest discrepancy between the DL model and TCR-2 was for the PRD region, where the
variations in the the DL-based NO$_x$ emissions did not match that in TCR-2 before nor after the CNY. For example, TCR-2
exhibited large variations in the emissions, which the DL model does not capture. As discussed above, the DL model has
difficulty reproducing the TCR-2 emissions in the Pearl River Delta throughout 2019 even after Stage 1 of the training, when
the model is trained solely on TCR-2 data, so the discrepancy between TCR-2 and the DL model in the signal of the CNY in
the PRD emissions is consistent with the results shown in Figure 5.

To validate the faster post-holiday recovery in the DL-based NO$_x$ emissions we use the C-index from the Baidu "Qianxi"
data, which is also shown in Figure 8. During the 2019 CNY period, the average C-index over the whole country rapidly
decreased by 10% from 4.5 to 4.1. It should be noted that the relationship between the C-index and NO$_x$ emissions is not linear,
as a 20% decrease in NO$_x$ emissions does not necessarily correspond to a comparable decrease in the C-index. However, as
a measure of human activity and a proxy for NO$_x$ emissions, the timing of the recovery in the C-index could provide useful
information to evaluate the performance of the models in capturing the relative variations in NO$_x$ emissions, especially from
transportation source sectors. As shown in Figure 8, the faster recovery of the DL model is consistent with the C-index. This is
particularly evident for China, JJJ, and the YRD. Figure 9 shows the estimated time (in days) for the DL model, the TCR-2 NO$_x$
emissions, and the C-index to recover to 50% of the averaged values five days before the CNY for China, the three Chinese
megaregions, and all Chinese provinces. For all provinces, the recovery of the C-index took $9.5 \pm 5.2$ days after the CNY (Figure
9b). In comparison, the recovery of the NO$_x$ emissions in the DL model and TCR-2 took $14.4 \pm 8.4$ days and $23.0 \pm 11.1$ days,
respectively. For JJJ and the YRD, the DL model suggested a recovery time of 15 days and 16 days, respectively, whereas the
C-index recovery took 9 days (Figure 9a). For the PRD, the DL model recovered after 15 days, whereas the C-index recovered
after 18 days.

The validation of the DL estimates here provide insights about the limitations of the satellite-derived NO$_x$ emissions. Insuf-
ficient space-based observational constraints can limit the representation of short-term changes in NO$_x$ emissions. The OMI



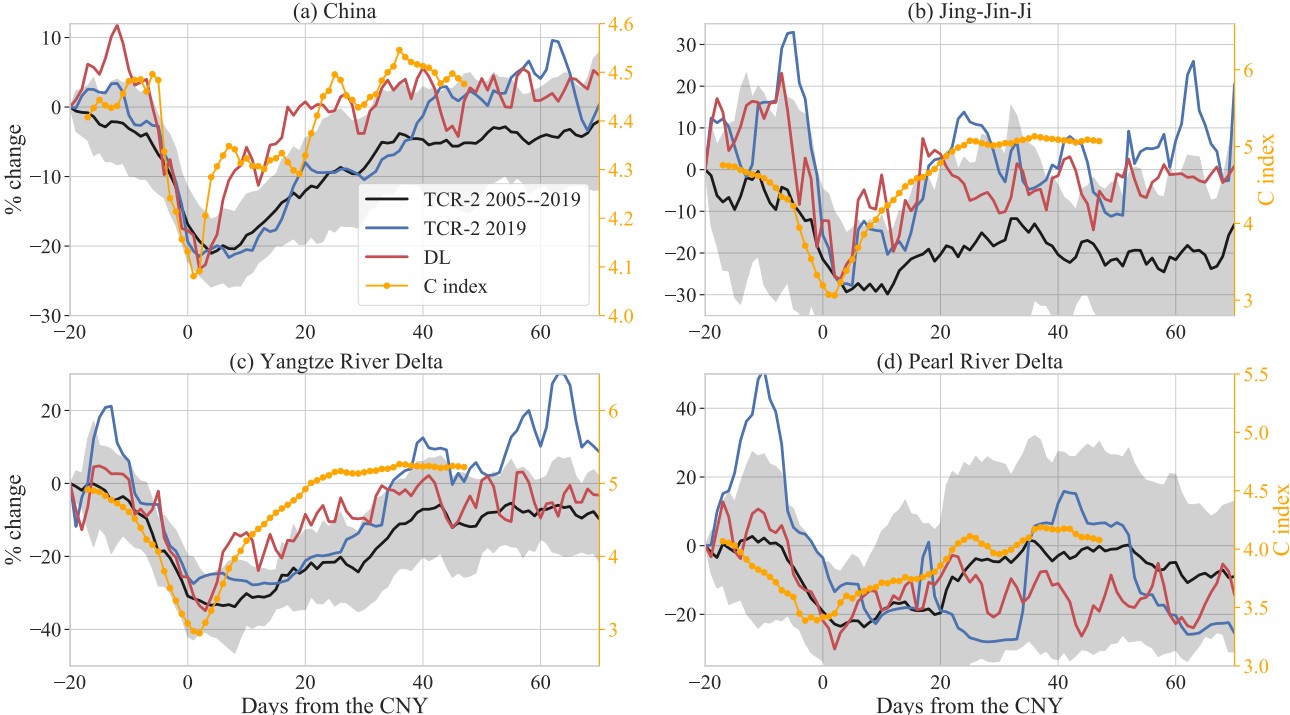

**Figure 8.** Time series of the percentage change in Chinese NO$_x$ emissions and the C-index Baidu mobile data as a function of days from the CNY. The differences are plotted relative to 20 days before the 2019 CNY. Shown are time series comparisons for (a) China, (b) the Jing-Jin-Ji region, (c) the Yangtze River Delta region, and (d) the Pearl River Delta region. The black line represents the TCR-2 mean for 2005–2019. The C-index is smoothed by a 7-day window to remove weekly variability.

observational coverage is limited especially during winter over China due to cloudy or rainy conditions, and the relatively large retrieval uncertainty could prevent rapid a posteriori emission changes in the top-down analysis. This limitation could be mitigated by further optimization of the background error covariances in the top-down analysis to better reflect individual
measurements. In addition, the use of more dense and accurate observations, such as from TROPOMI (see Section 3.3) could provide an improved representation of daily emission changes.



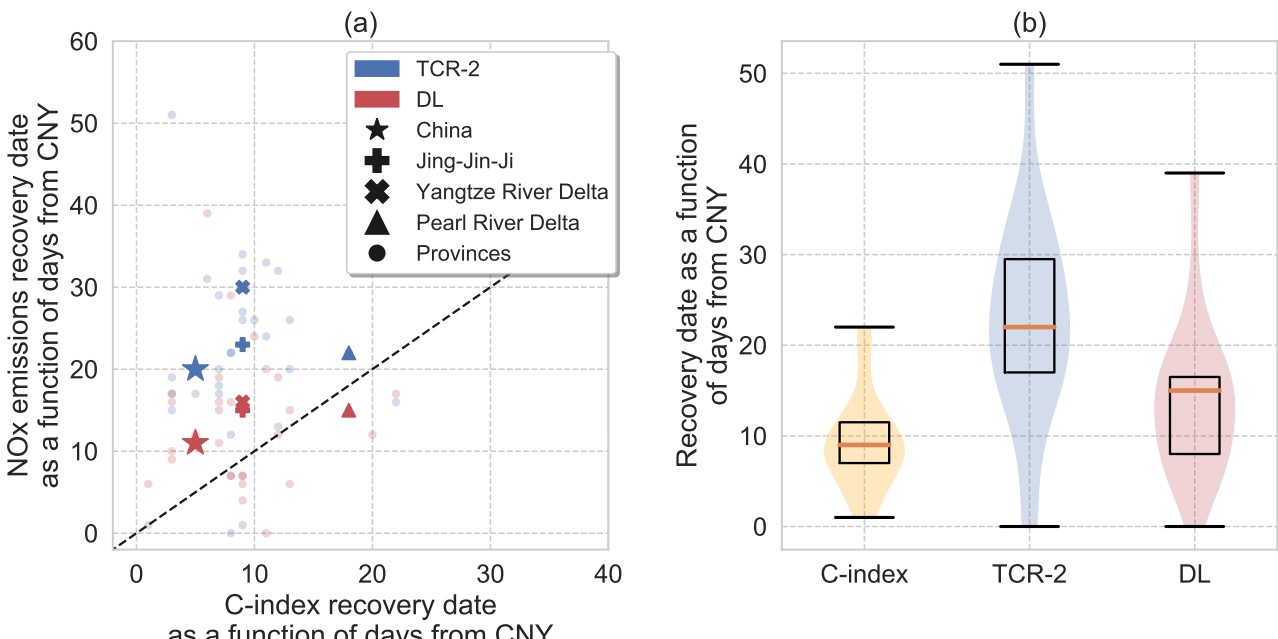

**Figure 9.** Comparison of the timing of the recovery of the NO$_x$ emissions and Baidu C-index data back to 50% of the averaged values five days before the CNY in 2019. (a) The scatter plot of NO$_x$ emission recovery dates versus the C-index recovery dates. Each circle represents a province and special markers correspond to the larger regions as indicated by the legend. (b) Box whisker plots and the normalized distribution of the recovery dates for the C-index, TCR-2 NO$_x$ emissions, and the DL-based NO$_x$ emissions, calculated based on all provinces in China.



### 3.3 Analysis of the 2020 COVID-19 pandemic period

Since the COVID-19 pandemic lockdown led to a significant and unexpected perturbation to human activity, we examine the ability of the DL model to quantify changes in $NO_x$ emissions during the lockdown period in China in 2020. Here we use the TROPOMI-based higher-resolution analysis using TCR-2 (hereafter referred to as T213 data) as an independent data set for the evaluation of the DL model. The standard OMI-based TCR-2 data product is hereafter referred to as T106 data. The T213 data were used to study detailed spatial and temporal changes in $NO_x$ emissions during the COVID lockdowns (Miyazaki et al., 2020b, 2021). The T213 product is expected to better capture variability in $NO_x$ emissions, compared to the T106 product, because of the greater observational constraints from TROPOMI and the higher spatial resolution and coverage. The time series of the relative changes in $NO_x$ emissions around the 2020 CNY are shown in Figure 10. We choose the reference date to be 10 days before the 2020 CNY to avoid spin-up issues in the T213 data. On the national scale, all three $NO_x$ emission estimates decreased at roughly the same rate until about 10 days after the 2020 CNY, after which the DL-based emission began increasing. The T106 and T213 $NO_x$ emissions continued to decrease for about an additional 10 days after CNY, with the T213 data suggesting a smaller over all reduction than T106. All three $NO_x$ emission estimates suggested that a full recovery took 60 days after CNY. However, of the three, the DL-based estimates suggested the smallest overall reduction in Chinese $NO_x$ emissions. For JJJ, the DL-based and T106 emissions estimates were fairly consistent, but the T213 estimates suggested a faster recovery.

Comparison of the $NO_x$ emission time series with the C-index in Figure 10 shows distinct differences in the timing of the minimum in the data after the 2020 CNY. As noted above, we do not expect there to be a linear relationship between the C-index and the $NO_x$ emissions, but since the reduction in emissions is in part driven by the lockdown, we anticipate that the minimum in the migration data should closely correspond to the minimum in emissions. For China, as listed in Table 3, the C-index reached a minimum 14 days after the CNY, whereas the DL model, the T213 data, and the T106 data reached a minimum within 13, 21, and 23 days, respectively. The DL model reproduced well the timing of the minimum for JJJ and the YRD. However, it significantly underestimated the timing for PRD (3 days for the DL model compared to 12 days for the C-index). We find that the timing of the minima in the T213 emissions more closely match that of the C-index than T106, which suggested delayed minima.

The timing of the recovery of the post-holiday reductions to 50% of the averaged values 5 days before the holiday for the C-index and the estimated $NO_x$ emissions are listed in Table 4. For China, the C-index took 31 days to recover in 2020, whereas the DL model and the T213 data took 31 days and 37 days, respectively, to recover. The recovery time in T106 data was 41 days. For JJJ, the time required for the 50% recovery was 34 days for the C-index, whereas for the DL model, the T213 data, and the T106 data it was 34 days, 36 days, and 31 days, respectively. For the YRD, both reanalysis data products took over 40 days to recover, which is more than 20 days longer than the C-index and more than 10 days longer than the DL model. In the PRD, all of the estimated $NO_x$ emissions took more than 20 days longer to recover than the C-index.

The spatial distribution of the DL-estimated changes in Chinese $NO_x$ emission 20–30 days after the 2020 CNY, relative to 10–20 days prior to the CNY, are shown in Figure 11. The DL analysis shows over 30% reduction in $NO_x$ emissions 20–30



**Table 3.** Timing (in days) of the minimum in the migration data and $NO_x$ emissions during 30 days after the 2020 CNY.

| Regions | C-Index | DL Model | T213 | T106 |
|---------|---------|----------|------|------|
| China | 14 | 13 | 21 | 23 |
| Jing-Jin-Ji | 12 | 13 | 20 | 28 |
| Yangtze River Delta | 12 | 13 | 17 | 27 |
| Pearl River Delta | 12 | 3 | 13 | 20 |

**Table 4.** Time (in days) for the migration data and $NO_x$ emissions to recover to 50% of the pre-CNY level in 2020.

| Regions | C-Index | DL Model | T213 | T106 |
|---------|---------|----------|------|------|
| China | 34 | 32 | 38 | 41 |
| Jing-Jin-Ji | 37 | 33 | 37 | 31 |
| Yangtze River Delta | 25 | 32 | 47 | 46 |
| Pearl River Delta | 27 | 24 | 65 | 52 |

days after the 2020 CNY for the heavily polluted regions in northern and eastern China. In JJJ, the DL analysis suggested a maximum reduction in $NO_x$ emissions of 42% during the lockdown period, whereas the T106 and T213 data indicated a maximum reduction of 49% and 41%, respectively. Using a regional model to assimilate the MEE data to estimate $NO_x$ emissions for January–March 2020, Feng et al. (2020) estimated a reduction of 42% in $NO_x$ emissions for JJJ. For the YRD,
290  the DL model and the T106 data suggested comparable reductions of 30% and 31%, respectively, which were roughly 10% smaller than the reductions in the T213 data and in Feng et al. (2020) of 40% and 41%, respectively. The comparison here shows that the impact of the lockdown on $NO_x$ emissions in the higher resolution T213 data, in contrast to the T106 data, is more consistent with Feng et al. (2020). This confirms that the emission analysis at T106 resolution (1.1°) constrained by the OMI measurements may not provide sufficient information to capture rapid regional variations in $NO_x$ emissions. The
295  four analyses exhibited the largest disagreement in the PRD, where Feng et al. (2020) estimated a 50% maximum reduction in emissions, whereas the T106 and T213 data suggested maximum reductions of 39% and 35%, respectively. The DL model significantly underestimated the emission reduction in the region, with a maximum reduction of 24%. Overall, the comparison with the migration data and with the Feng et al. (2020) results indicate that, with the exception of the PRD, integrating the MEE and TCR-2 data results in an improved relationship between surface $NO_2$ concentrations and $NO_x$ emissions.



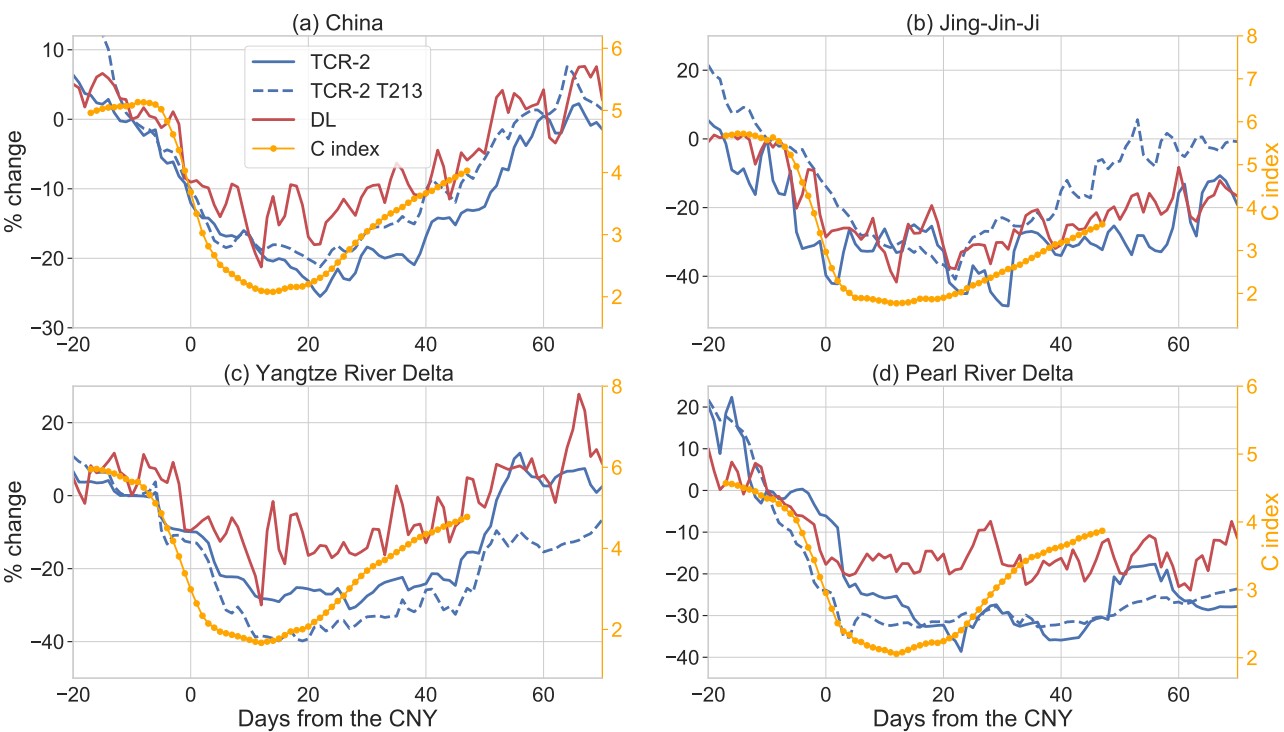

**Figure 10.** Time series of percentage changes in $NO_x$ emissions and the C-index Baidu mobility data, relative to 10 days before the 2020 CNY, as a function of days from the CNY. Shown are the time series for (a) China, (b) the Jing-Jin-Ji region, (c) the Yangtze River Delta, and (d) the Pearl River Delta.

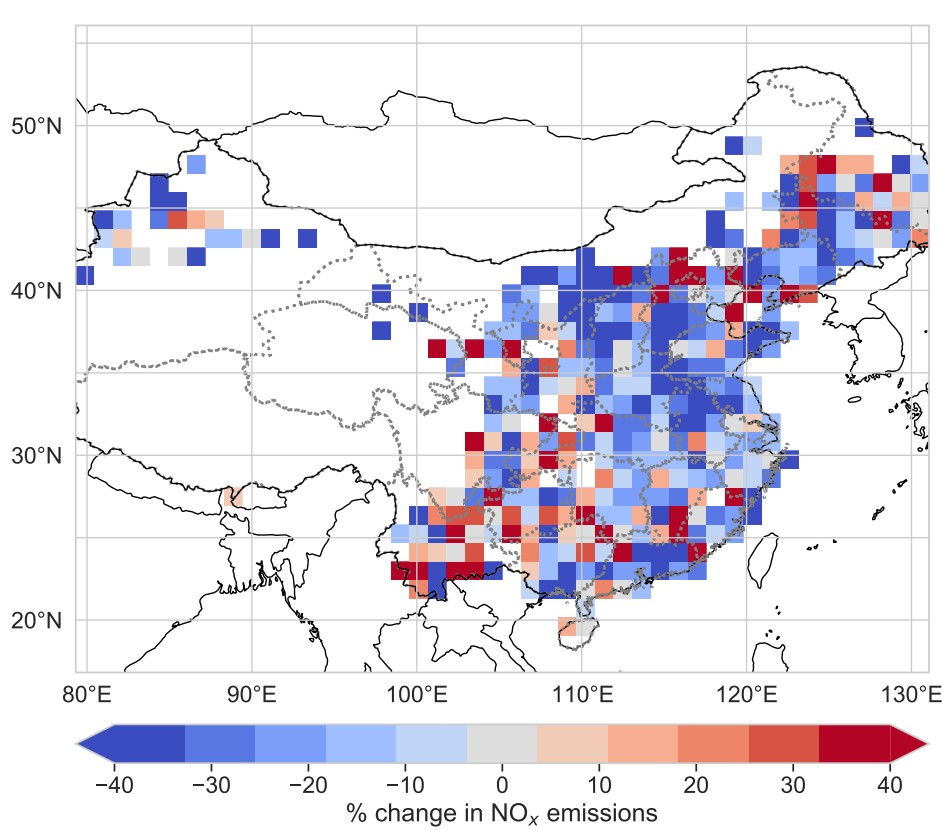

**Figure 11.** DL-estimated percent changes in Chinese $NO_x$ emissions averaged 20–30 days after the 2020 CNY compared to 20–10 days before the 2020 CNY. Results for grid boxes with $NO_x$ emissions less than $1 \times 10^{-11}$ kgN m$^{-2}$ s$^{-1}$ are not shown.



## 4 Conclusions

We developed a DL model to estimate Chinese $NO_x$ emissions using surface $NO_2$ concentration and meteorological predictors, based on the integration of MEE in situ $NO_2$ observations with TCR-2 $NO_2$ chemical reanalysis data. To that end, we applied a multi-stage training strategy for the transfer learning of the chemical relationship between surface $NO_2$ and $NO_x$ emissions in the TCR-2 data set. We found that the integration of the MEE in situ data with TCR-2 suggested $NO_x$ emissions from China for 2019 to be 19.4 Tg NO, which is consistent with the 18.5±3.9 Tg NO estimated by TCR-2. The DL model and TCR-2 were both consistent with the suggested Chinese source of 20.9 Tg NO in the MEIC inventory (Zheng et al., 2021). For the JJJ and YRD megaregions in China, the DL-based $NO_x$ emissions were higher than TCR-2 by 13.8% and 10.0%, respectively. The DL model particularly increased emissions in the less densely populated provinces, where the MEE observations indicated higher surface $NO_2$ abundances than in TCR-2. He et al. (2022) suggested that inversions using satellite observations to estimate $NO_x$ emissions have the potential to blend information from background $NO_x$ and surface emissions at coarse spatial resolutions. It is possible that much higher resolution than the $1.1° \times 1.1°$ of TCR-2 is needed for the satellite-based assimilation system to capture the surface $NO_2$ signal in these less densely populated provinces. We also found that the DL model could not reproduce the TCR-2 relationship between $NO_x$ emissions and surface $NO_2$ in the PRD, and integration of the MEE data resulted in large adjustments in the $NO_x$ emissions in the region. During the monsoon season, southern China experiences heavy and frequent rainfall, and the mountainous topography of the PRD and its proximity of the ocean could make it challenging for TCR-2 and the DL model to accurately account for the influence on surface $NO_2$ of the complex meteorology in the region at a resolution of $1.1° \times 1.1°$.

Analysis of the DL-based $NO_x$ emissions focused around the CNY holiday in 2019 showed a faster recovery of the Chinese $NO_x$ emissions after the 2019 CNY, which was consistent with the Baidu "Qianxi" mobile data (Kraemer et al., 2020; Zhang et al., 2021). During the 2020 lockdown period for COVID-19 pandemic, the DL model estimated maximum reductions in $NO_x$ emissions 42% for JJJ and 30% for the YRD. These estimates were consistent with the TCR-2 T106 data (with reductions of 49% for JJJ and 31% for the YRD), the high resolution TCR-2 T213 data (with reductions of 41% for JJJ and 40% for the YRD), and with Feng et al. (2020) (who estimated reductions of 42% for JJJ and 41% for the YRD). For the PRD, the DL model estimated a significantly smaller maximum reduction in $NO_x$ emissions of 24%, which is likely due to the model bias in the region.

The analysis during the 2020 lockdown period showed that the DL model has the ability to extrapolate outside the regime of the training data set. Our results showed the potential of this DL model as a good complementary tool for conventional data assimilation approaches. The flexibility of the model is such that it could be adapted to provide near-real time estimates of $NO_x$ emissions for air quality forecasts and chemical reanalysis systems. The high computational efficiency of the DL model in integrating large amounts of observational data from multiple sources would be advantageous in the emerging era with geostationary satellites that will significantly enhance observational coverage for air quality applications.





*Code and data availability.* The code for this study can be found at https://github.com/tailonghe/Unet_Chinese_NOx. The TCR-2 data could be accessed from https://tes.jpl.nasa.gov/tes/chemical-reanalysis/products/monthly-mean. The ERA5 climate reanalysis data is available from ECWMF https://www.ecmwf.int/en/forecasts/datasets/reanalysis-datasets/era5/. The MEE in situ observations were originally

downloaded from the MEE website (http://106.37.208.233:20035, last access: 18 February 2022) and were processed and archived at https://quotsoft.net/air by (last access: 18 February 2022). The Baidu "Qianxi" mobile data was originally retrieved from the Baidu "Qianxi" website (http://qianxi.baidu.com/, last access: December 2021) and processed in Zhang et al. (2021).

*Author contributions.* T.L.H. and D.B.A.J. designed the research study; T.L.H. built and trained the model; T.L.H. and D.B.A.J. performed research and analyzed results; T.L.H., D.B.A.J., K.M., K.W.B. and Z.J. contributed to revising and editing the manuscript.

*Competing interests.* The authors report no competing interests.

*Acknowledgements.* This work was supported by the Natural Sciences and Engineering Research Council of Canada. Computations were performed on the Graham supercomputer of Compute Ontario and Compute Canada. Part of this work was conducted at the Jet Propulsion Laboratory, California Institute of Technology, under contract with the National Aeronautics and Space Administration (NASA).



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
