# Peer review of "Inverse modeling of Chinese $NO_{\rm x}$ emissions using deep learning: Integrating in situ observations with a satellite-based chemical reanalysis"

_Atmospheric Chemistry and Physics, 2022_

## Author Response (AR1)

We thank the two reviewers for the thoughtful reviews and the suggestions on the paper. The manuscript has been modified based on the comments. Our response to the reviewers' comments is color-coded in **red** below.

**Reviewer #1:**

Figure 1 shows that the density of MEE networks station is highly inhomogeneous. When using nearest neighbor interpolation, Stations in less dense areas will have more spatial influence in the NO2 surface concentration product presented as input of the DL model. In less dense areas, the effective resolution of the product is not expected to be at 1.1° as in dense areas. Figure 7 (d) shows good correlation between DL minus TCR-2 emission difference and the density of networks as deduced from Figure 1. Have you investigated the role of the network density on the DL predictions through sensitivity tests (by lowering the number of stations used in dense areas for example) ?

> We thank the reviewer for raising this point. We did not conduct a sensitivity test to investigate that. In a previous study (Han et al., 2022) about predicting CO, we looked into the impact of observational coverage and density by removing 10% of the grid-based observations from the training of a similar DL model and evaluated the model performance over those grids. We did not find noticeable degradation of the model performance over the regions where observations were artificially removed from training.

> However, it is worth mentioning that the reviewer is correct here that low station densities could lead to representation errors in the grid-averaged observations. Discussion has been added to the main text to emphasize this point.

> Moreover, when analyzing the results, we focused on the three metropolitan regions, where the density of MEE network stations is high, which will mitigate the impact of the coarse station density on our conclusions.

It would be good to see in an example at what TCR-2 surface NO2 concentration, MEE network surface NO2 concentration and TCR-2 NOx emission field look like. It would help assess qualitatively the additional information the MEE network brings.

> We added Figure 2 to the paper to show the distributions of the TCR-2 and MEE surface NO2.

‣ Line 134: can the length of the latent vector be mentioned ?

> This information is now added to the text.

‣ Line 135 : the description of how the information is conveyed through the LSTM cells could be improved. From Figure 2, one could understand that a latent vector is the input

of a first LSTM and that the output of the first LSTM cell serve as the input of the second LSTM. But with information contained in Table 1, it seems that the input of the DL model is a sequence of 2 timesteps (T and T-1 day) * 35 height pixels * 46 width pixels * 9 channels (9 variables ) and that data for T is the input of the first LSTM cell and data for T-1 days is the input of a second cell. If it is the case, it should be described more clearly in the article.

We thank the reviewer for pointing out this confusion. Instead of treating the input parameters as two time steps, we just treat information from different days as different channels in the input layer. Moreover, considering the short lifetime of surface NOx (typically several hours), we didn't include meteorological variables from day $t$-1. Namely, only meteorological information from day $t$ is used as predictors. We did include surface NO2 from days $t$ and $t$-1 day for better prediction of NOx emissions. So the full set of predictors is meteorological variables and surface NO2 from day $t$ plus surface NO2 from day $t$-1. We have added text explaining this.

▸ Table 1 : it is not stated that meteorological field include 2 timesteps.

We now explain in Table 1 that the NO2 concentrations are input from two different days. We do not use the meteorological fields at multiple time steps.

▸ Figure 2 : it could be mentioned that the DL model for stage 1 uses only 8 channels and 9 channels for stage 2.

In stage 2 training, instead of treating MEE surface NO2 as a new predictor, we integrate the information by filling the data gaps of the MEE surface NO2 with values from the TCR-2 reanalysis. This is now clarified in the Figure caption.

▸ Line 167, 'year 2020 is anomalous compared to the training set with normal years'. As industrial activity in China has dramatically increased from the 2000s in China, how is the level of NOx emission in 2020 compared to 2014 or 2005 ? It is maybe not so anomalous compared to the first years of the training set.

The TCR-2 NOx data we have are from the first 115 days of 2020. The mean Chinese NOx emissions for that period in 2020 were 10% and 23% lower than the same period in 2005 and 2014, respectively, and the change in emissions was faster than at any point in the training set. This information is added to the text.

▸ Line 274 : It is stated that the minimum for PRD for DL model is reached is 3days after CNY compared to 12 days. Actually, I can see on the red curve 3 minima (at 3, 13 and 17 days after CNY) in the 0-20 days after CNY period, so I don't' necessarily agree with the statement. The signal from DL seems very noisy compared to the other

products. Filtering it out with a moving average window would lead to different conclusions for PRD and highlight a plateau more than a global minimum on the curve.

> The reviewer is correct, the estimate of the PRD signal is less reliable because the signal is essentially flat. As we noted, the DL model has much less skill with the PRD. We have added text to make it clear that the timing of the minimum for the PRD is not well defined, as the signal is noisy and exhibits a fairly broad minimum.

‣ Line 25 : a word seems missing between 'changes' and 'the N02 column'

> Thank you. The typo is fixed in the paper now.

‣ Table 1 : is it 'unit before scaling' instead of 'unit after scaling' ? because after scaling, there is no more unit.

> Thank you for pointing this out. The original units of the raw data are different from the units in the table. We have modified the table to resolve this confusion.

**Reviewer #2:**

Section 3.2. validation using C-index. I would suggest clarifying the driver of the C-index change before the analysis. It's easy to understand the relationship between C-index and NOx emissions is not linear since they are potentially driven by different activities. As far as my understanding, the change of C-index is closely related to mobile emissions. However, the urban NOx emissions have multiple sources. Can we assume the differences in drivers are the reason for different recovery time? If so, do the differences suggest the mobile emissions recover quicker than other sources?

Additionally, the comparison among regions shows very diverse patterns. It seems the consistency for JJJ is significantly worse than other regions. Any insight about this? The similar questions are applied to section 3.3. I believe it will be worth to investigate the driver for the differences, since the authors try to us C-index to validate the method.

> C-index measures the intra-city transport level, including shopping, economy activities, etc. Therefore, C-index is a general proxy for human activities instead of only mobile emissions. Zhang et al. (2021), which was referenced in Section 2.3, examined the relationship between the C-index and NOx, and showed that over 40% of variance of emission-based NO2 reductions in China could be explained by the C-index. For some megacities in southern China, the variance

explained by the C-index was as large as 70%. We have added text explaining this. It would be beyond the scope of this manuscript to extend the work of Zhang et al. (2021) and investigate the drivers of the variations in the C-index.

‣ Page 1, line 21. NOx also has natural sources from the surface.

Thank you. The sentence is revised to include natural NOx sources.

‣ Page 3, line 58. Please try to define "two-stage transfer learning strategy" before using them.

A sentence is added to define the term.

‣ section 2.1. Please clarify which OMI and TROPOMI products and data version are used.

The data products and versions are added in the text now.

‣ Page 6. Line 119. What is the magnitude of the correlation? It is good to use a few sentences to explain the reason behind the correlation here.

We had added text to the manuscript giving the correlation.

*References:*

Han, W., He, T.-L., Tang, Z., Wang, M., Jones, D., and Jiang, Z.: A comparative analysis for a deep learning model (hyDL-CO v1.0) and Kalman filter to predict CO concentrations in China, Geosci. Model Dev., 15, 4225–4237, https://doi.org/10.5194/gmd-15-4225-2022, 2022.

Yuxiang Zhang, Haixu Bo, Zhe Jiang, Yu Wang, Yunfei Fu, Bingwei Cao, Xuewen Wang, Jiaqi Chen, Rui Li, Untangling the contributions of meteorological conditions and human mobility to tropospheric NO2 in Chinese mainland during the COVID-19 pandemic in early 2020, National Science Review, Volume 8, Issue 11, November 2021, nwab061, https://doi.org/10.1093/nsr/nwab061